# Mitogen-Activated Protein Kinase Is Involved in Salt Stress Response in Tomato (*Solanum lycopersicum*) Seedlings

**DOI:** 10.3390/ijms23147645

**Published:** 2022-07-11

**Authors:** Lijuan Wei, Li Feng, Yayu Liu, Weibiao Liao

**Affiliations:** College of Horticulture, Gansu Agricultural University, 1 Yinmen Village, Anning District, Lanzhou 730070, China; wlj920229@163.com (L.W.); feng1654114573@163.com (L.F.); liuyayu199809@163.com (Y.L.)

**Keywords:** mitogen-activated protein kinase, salt toxicity, seedling growth, plant immunity, hormone balance, RNA sequencing

## Abstract

Salt stress impairs plant growth and development, thereby causing low yield and inferior quality of crops. In this study, tomato (*Solanum lycopersicum* L. ‘Micro-Tom’) seedlings treated with different concentrations of sodium chloride (NaCl) were investigated in terms of decreased plant height, stem diameter, dry weight, fresh weight, leaves relative water content and root activity. To reveal the response mechanism of tomato seedlings to salt stress, the transcriptome of tomato leaves was conducted. A total of 6589 differentially expressed genes (DEGs) were identified and classified into different metabolic pathways, especially photosynthesis, carbon metabolism, biosynthesis of amino acids and mitogen-activated protein kinase (MAPK) signaling pathway. Of these, approximately 42 DEGs were enriched in the MAPK signaling pathway, most of which mainly included plant hormone, hydrogen peroxide (H_2_O_2_), wounding and pathogen infection signaling pathways. To further explore the roles of MAPK under salt stress, MAPK phosphorylation inhibitor SB203580 (SB) was applied. We found that SB further decreased endogenous jasmonic acid, abscisic acid and ethylene levels under salt stress condition. Additionally, in comparison with NaCl treatment alone, SB + NaCl treatment reduced the content of O^2−^ and H_2_O_2_ and the activities of antioxidant enzyme and downregulated the expression levels of genes related to pathogen infection. Together, the results revealed that MAPK might be involved in the salinity response of tomato seedlings by regulating hormone balance, ROS metabolism, antioxidant capacity and plant immunity.

## 1. Introduction

Nowadays, water scarcity and quality degradation are major constraints on agricultural development. In addition, the introduction of irrigated agriculture in arid and semiarid regions has caused secondary soil salinization. Facing salt stress, plants adapt through different morphological and cellular responses [1]. Salinity could influence more than 6% of the world’s total land area (about 800 million hectares of land worldwide) [2]. Meanwhile, high salinity is commonly due to overabsorption of Na^+^ and Cl^−^ in the soil solution and further leads to hyperosmotic and hyperionic conditions, which inhibit the ability of plants to uptake water and nutrients from the soil [3]. Plants must be able to survive under stress. Thus, with time, plants had to evolve different mechanisms to adapt to high-salinity conditions.

Salt stress is usually accompanied by ionic stress, osmotic stress and oxidative stress due to the overproduction of reactive oxygen species (ROS) in plants, causing the oxidation of protein, membrane lipids and nucleic acids, and inhibiting plant growth and development [4,5]. Additionally, under salt stress, plants have to extensively adjust various physiological and biochemical processes, including ion and osmotic homeostasis, as well as stress damage control and repair [2]. Besides, it impedes cell division and expansion, thus inhibiting seed germination, seedling growth, plant life and crop productivity [5,6]. Meanwhile, under stress, plant genes responsible for plant resistance are activated resulting in the production of signaling pathways, which could enhance plant tolerance to salt stress. For example, Sun et al. [7] indicated that TaZFP1-mediated salt tolerance is ascribed to the regulation of gene functions related to photosynthesis, ROS homeostasis and osmolytes metabolism. Therefore, it is important to evaluate plant response and explore the activation of signaling pathways under salt stress so that the production of more tolerant plants may be possible. 

Mitogen-activated protein kinases (MAPKs) are the intracellular signal transducers, which transduce extracellular stimuli into plant cells via three layers of protein kinases, including MAPK kinase kinases (MAPKKKs), MAPK kinases (MAPKKs), and MAPKs. When plants respond to stimulants, MAPKKKs phosphorylate and thus activate MAPKKs, which in turn phosphorylate MAPKs. Phosphorylated MAPKs translocate from the cytoplasm to the nucleus, and regulate gene expression, thereby modulating the growth and development of plants [8]. During stress, different signaling pathways are activated, especially the activation of MAPK signaling cascade. Heinrich et al. [9] found that, in *Nicotiana attenuata*, two MAPKs, wounding-induced protein kinase (NaWIPK) and salicylic acid-induced protein kinase (NaSIPK), are remarkedly activated in response to wounding. The activation of different MAPK components and the subsequent expression of stress genes may affect plant response to stress.

Tomato (*Solanum lycopersicum* L. ‘Micro-Tom’) is a wild relative of the cultivated species. Moreover, the tomato crop frequently experiences various adverse conditions in some parts of the world. Tomato is a salt-sensitive plant, and it can be severely affected by salinity at all stages after its plantation. Thus, the reproductive behavior of plants under various extreme environments needs to be extensively studied [10]. Furthermore, there are a lot of missing links between tomato seedling growth and MAPKs under salt stress. Therefore, it is important to discover the possible effects of various MAPK signaling on growth and development in tomato seedlings under sodium chloride (NaCl) stress and their possible mechanisms. The objectives of the present study were to investigate the impact of NaCl as moderate salt stress on the physio-morphological attributes and the growth of tomato cv. ‘Micro-Tom’ seedlings. Then, RNA sequencing (RNA-seq) technology was conducted to identify the candidate genes associated with the MAPK signaling pathway in salt-stressed tomato seedlings and a control. Last, according to RNA-seq results associated with the MAPK signaling pathway, we mainly focused on the change in the levels of endogenous hormones, ROS metabolism, antioxidant ability and defense response to pathogens after supplying MAPK inhibitors SB203580 in tomato seedlings under salt stress. 

## 2. Results

### 2.1. Effects of Various Concentrations of NaCl on Seedlings Developmen

To explore the inhibitive effect of NaCl on tomato growth, the seedlings were treated with different concentrations of NaCl (0, 50, 100, 150, and 200 mM). As shown in Figure 1, the plant height, stem diameter, dry weight and fresh weight showed a reducing trend at varying degrees with increasing NaCl concentration. Among them, there was no difference between the control and 50 mM NaCl, whereas, compared with the control, the 100, 150 and 200 mM NaCl significantly decreased the plant height, stem diameter, dry weight and fresh weight (Figure 1A,B). Compared to the control, in the higher concentrations of NaCl (100, 150 and 200 mM) treatments, the plant height decreased by 13.83%, 19.99% and 21.22%, respectively; the stem diameter decreased by 10.42%, 18.09%, and 22.69%, respectively; the dry weight decreased by 23.53%, 29.41%, and 32.35%, respectively; and the fresh weight decreased by 25.00%, 38.67%, and 44.35%, respectively (Figure 1). In addition, the leaf relative water content (LWC) and root activity also showed a similar trend (Figure 1C). The higher concentrations of NaCl significantly decreased LWC and root activity, and 50 mM NaCl had no significant inhibitive effect compared to the control. Moreover, in comparison with the control, tomato leaves became slightly curled and yellowed in 150 and 200 mM NaCl treatments (Figure 1D). These results suggested that NaCl might have a concentration-dependent effect on the growth and development of tomato seedlings and that 150 mM NaCl was the suitable concentration for moderate salt stress and was therefore used for the following experiments.

### 2.2. Identification and Functional Classification of Differentially Expressed Genes (DEGs) under Salt Stress

To better understand the basis of the molecular mechanisms during salt stress, the RNA-seq analysis was determined. RNA-seq obtained raw data from 44.6 to 55.3 million reads (Appendix A). After filtering, more than 47.32 million clean reads and 45.97 G clean bases were produced. The average percentage of Q20 and Q30 reached 97.69 and 93.36%, respectively. The GC (guanine + cytosine) contents of samples were about 42–43% and the GC content of each group remained basically stable and unchanged suggesting a higher quality of sequencing that could be applied to further analysis. To further study the metabolism pathway that occurs in tomato seedlings during salt stress, KEGG and gene ontology (GO) enrichment was used for analysis of the DEGs. As shown in Figure 2, in NaCl vs. control, a total of 6589 DEGs was obtained. The number of downregulated DEGs (3179) was more than that of upregulated DEGs (2917), which might be associated with various metabolic pathways (Figure 2A; Appendix A). As shown in Figure 2B, the GO database was used to categorize DEGs. In the GO enrichment analysis, the classification of DEGs was as follows: biological process (BP), cellular component (CC) and molecular function (MF). During the biological processes, 135 and 89 DEGs treated with NaCl were mainly concentrated in small molecule metabolic process (GO:0044281) and oxoacid metabolic process (GO:0043436), respectively. In the cellular component, the DEGs were mainly involved in thylakoid (GO:0009579, 42 DEGs) and photosystem (GO:0009521, 38 DEGs). During the molecular function, 102 DEGs and 66 DEGs were mainly enriched in cofactor binding (GO:0048037) and coenzyme binding (GO:0050662) (Figure 2B; Appendix A). Based on comparison with the KEGG database, a total of 1336 DEGs were assigned to 117 KEGG pathways (Appendix A). In Figure 2C, the top 20 KEGG pathways corresponding to the most abundant DEGs are shown. The results indicated that most of the DEGs were enriched in the photosynthesis, carbon metabolism, biosynthesis of amino acids and MAPK signaling pathway. Therefore, these results implied that salt toxicity may affect these metabolism pathways, thereby regulating the growth and development of tomato seedlings. Among these pathways, the MAPK signaling pathway was selected for subsequent analysis.

### 2.3. Analysis and Confirmation of DEGs Involved in MAPK Signaling Pathway under Salt Stress

According to the KEGG results, 42 significantly enriched DEGs were in the MAPK signaling pathway (sly04016), most of which respond to salt stress (Figure 3 and Appendix A). The MAPK signaling pathway was mainly involved in plant hormone, hydrogen peroxide (H_2_O_2_), wounding and pathogen infection signaling pathways. Plant hormones mainly include ethylene (ETH), abscisic acid (ABA) and jasmonic acid (JA). There were 11 DEGs identified in the ETH signaling pathway. *Ethylene receptor* (*ETR*), EIN/EIL, *ethylene response factor* (*ERF*) and *chitinase* (*CHI*) are key genes involved in the ETH signal transduction pathway. Two *CHI* genes (*CHI3* and *CHI9*), two *EIL* genes (*EIL3* and *EIL4*), *ETR* gene (*ETR6*) and *ERF* gene (*ERFC.3*) were significantly upregulated in NaCl treatment. *EIN3-3like* and *ACS6* related to ETH synthesis were downregulated in the NaCl treatment compared to the control. In the ABA signaling pathway, the expression levels of the *serine/threonine-protein kinase gene* (*SRK2I*) were significantly upregulated by NaCl. Meanwhile, three *protein phosphatase 2C* (*PP2C*) genes, *PP2C*, *PP2C 53* and *PP2C 37*, and ABA receptor *PYL3* were downregulated, while *PP2C 51-like* and *ABA receptor 3-like* were upregulated under salt stress compared to the control. Meanwhile, a key gene, *MYC2* related to JA signaling transudation was significantly downregulated in tomato seedlings during salt stress. There were 9 DEGs that were significantly enriched in the H_2_O_2_ pathway. Among them, a catalase gene, *cat1*, related to H_2_O_2_ reduction, and a mitogen-activated protein kinase gene (*MPK9*) were upregulated, whereas, pathogenesis-related leaf protein (*PR1* and *SlPR4*), WRKY transcription factor (*SlWRKY22* and *SlWRKY22* like) and *CAT2* were downregulated under salt stress, then regulating H_2_O_2_ production. Calmodulin (CaM), MPK8, and respiratory burst oxidase homolog protein (RbohD) are key regulators in wounding and are helpful in the maintenance of the homeostasis of ROS. Here, *CML30*, *RbohD* (LOC101246440) and *MAPK16* were downregulated by salt stress. In addition, EPIDERMAL PATTERNING FACTOR 1 (*EPF1*) was upregulated by salt stress, thereby regulating stomatal development. In the pathogen infection pathways, WRKY transcription factor (*WRKY 33B* and *SlWRKY33A*), two MKS1 genes (*LOC101244245* and *LOC104647127*), LRR receptor-like serine/threonine-protein kinase *FLS2* and probable LRR receptor-like serine/threonine-protein kinase *At3g47570* were downregulated, while the expression levels of probable LRR receptor-like serine/threonine-protein kinase *At3g47570* (*LOC101260980*) and transcription factor VIP1 (*tfVIP1*) were significantly upregulated under salt stress compared to the control. These results highlighted various functions of DEGs enriched in the MAPK signaling pathway, which might participate in salt stress response in tomato seedlings.

To determine the reliability of our transcriptome data, the expression of 13 DEGs related to MAPK signaling under salt stress was conducted (Figure 4). These DEGs included four ETH-related genes (*endochitinase*, *CHI9*, *CHI3*, *EIL3*), two ABA-related genes (*PP2C 37*, *PP2C 51 like*), a JA-related gene (*MYC2*), two H_2_O_2_-related genes (*cat1*, *MPK9*), two wounding-related genes (*CML30*, *MAPK16*) and two pathogen infection-related genes (*WRKY 33B* and *SlWRKY33A*). Compared with the control, the expression levels of *CHI9*, *EIL3*, *PP2C 51 like*, *cat1* and *MPK9* exhibited a significantly increased trend in NaCl treatment, while the expression levels of *endochitinase*, *PP2C 37*, *MYC2*, *MAPK16 and SlWRKY33A* were significantly decreased in NaCl treatment. These results indicated that the expression patterns of all genes analyzed by qRT-PCR were consistent with our transcriptome analysis, confirming the reliability of the RNA-seq data, and implying that the MAPK signaling was stimulated during salt stress at varying degrees.

### 2.4. Effect of MAPK Inhibitors SB203580 on the Levels of Endogenous Hormones under Salt Stress

To further explore the roles of MAPK in the process of salt stress, MAPK protein phosphorylation inhibitor SB was applied in our research. Since 42 DEGs were enriched in the MAPK signal pathway under salt stress and most of the DEGs were involved in plant hormones signal transduction, including ETH, ABA and JA (Figure 3), the content of three plant hormones were determined in different treatments (Figure 5). Compared with the control, ETH production and ABA content were significantly increased under salt stress, while JA content was opposite (Figure 5). However, the supplementation of the MAPK inhibitor SB significantly reversed the increasing effect of NaCl. Compared with NaCl treatment alone, NaCl + SB treatment increased the content of ETH and ABA by 20.84% and 38.92%, respectively. Meanwhile, in comparison with the control, the JA content was significantly decreased by NaCl treatment, whereas the reductive effect was promoted after adding MAPK inhibitor SB under salt stress. Additionally, compared with the control, SB treatment significantly reduced ETH production, ABA and JA content in tomato seedlings. Combined with the above results, it could be concluded that the change mode of endogenous plant hormones is due to the MAPK signaling pathway under salt stress, and MAPK has an essential role in regulating the levels of endogenous plant hormones when tomato seedlings responded to salt stress.

### 2.5. Effect of MAPK Inhibitors SB203580 on the Accumulation of Reactive Oxygen Species under Salt Stress

Among 42 DEGs enriched in the MAPK signal pathway, various DEGs were related to H_2_O_2_ production and ROS accumulation (Figure 3). To elucidate the relationship between MAPK and ROS accumulation under salt stress, the MAPK inhibitor SB was used, and then the H_2_O_2_ content and O^2−^ production rate were determined in different treatments (Figure 6). As shown in Figure 6, under salt stress, the H_2_O_2_ content and O^2−^ production rate showed an increasing trend compared to the control. Additionally, in comparison with NaCl treatment alone, NaCl + SB treatment reduced the H_2_O_2_ and O^2−^ overproduction, which showed a decrement of 17.64% and 22.09%, respectively. In SB treatment alone, the H_2_O_2_ content slightly decreased, whereas the O^2−^ production rate had no significant difference compared with the control. Thus, these results indicated that MAPK might participate in ROS over-accumulation induced by salt stress in tomato seedlings.

### 2.6. Effect of MAPK Inhibitors SB203580 on Activities of Antioxidant Enzyme under Salt Stress

Because various DEGs enriched in the MAPK signaling pathway were related to ROS accumulation and homeostasis, and wounding, the activities of antioxidant enzymes were determined after supplying MAPK inhibitors SB203580 to decipher the relationship of MAPK and antioxidant capacity in tomato seedlings under salt stress. As is shown in Figure 7, the activity of CAT significantly decreased, which showed a decrement of 35.09% in 150 mM NaCl treatment compared with the control, whereas the inhibitive effect of NaCl was weakened by adding SB. In addition, in comparison with the control, NaCl treatment significantly enhanced the SOD, POD and APX activities. However, NaCl + SB treatment decreased the activities of SOD, POD, and APX, which was 27.22%, 22.38%, and 13.28% lower than salt stress, respectively. In addition, in SB treatment alone, SOD, POD, and APX activities had a significantly decreasing trend compared with the control. Thus, these results indicated that MAPK have an important role in regulating the antioxidant capacity of tomato seedlings under salt stress.

### 2.7. Effect of MAPK Inhibitors SB203580 on the Expression Patterns of Genes Related to Defense Response for Pathogen under Salt Stress

Since various DEGs in the MAPK signal pathway were related to “pathogen infection”, MAPK inhibitor SB was added to determine the relative expression levels of genes related to pathogen infection (Figure 8). *MPK3* and *MPK4* act upstream of the transcription factors *WRKY 22* and *WRKY33* in the pathogen infection pathway, respectively. Compared with the control, *MPK3* and *MPK4* expression levels were downregulated under salt stress, which further significantly decreased after adding MAPK inhibitor SB, implying that MAPK have an essential role in defense response to pathogens under salt stress. Meanwhile, WRKY transcription factor, *SlWRKY 22*, *SlWRKY 33A* and *SlWRKY 33B* expression levels were significantly downregulated in NaCl + SB treatment in comparison with NaCl treatment alone, whereas the *SlWRKY 22* expression had no significant change. In addition, compared with the control, SB treatment alone also decreased the relative expression levels of WRKY transcription factors. MKS1, a MPK4-substrate, is required for basal resistance against pathogen infection. We found that NaCl treatment significantly inhibited the expression of two *MKS1* genes compared with the control, which further downregulated in NaCl + SB treatment. Therefore, these results suggested that salt stress decreased the resistance against pathogen infection of tomato seedlings, whereas the activation of MAPK might have a positive role in plant immunity to pathogen infection.

## 3. Discussion

Abiotic stresses have a detrimental effect on crop performance, and especially, soil salinity is a significant constraint on yield and quality of crop [2,11]. Salt stress impairs plant physiology at whole-plant and cellular and molecular levels, and at all developmental stages from germination to senescence [1,12]. For example, water stress impacts on the physiological behavior of young seedlings of *Acacia arabica*, thereby enhancing sensitivity to progressive soil drying [13]. In particular, the young seedling stage of plants is especially sensitive to adverse environmental conditions. In our study, the plant height, stem diameter, dry weight, fresh weight, LWC and root activity showed a significant reducing trend at varying degrees with increasing NaCl concentration (0, 50, 100, 150, and 200 mM) (Figure 1). Similarly, different types of salt concentrations (0, 120, 150 mM) remarkedly reduced the plant height, leaf area stem diameter, number of leaves and LWC of pearl millet in a dose-dependent manner [14]. In addition, Elsawy et al. [15] indicated that 200 mM NaCl treatment for 12 days could inhibit the plant height, relative growth rate and leaf area of *Egyptian barley*. Salt stress in soils, on the one hand, impairs the plant’s ability to take up water from the rhizosphere, leading to water limitation and growth inhibition. On the other hand, it is well known that salt toxicity inhibits shoot growth by reducing the initiation, expansion and yellowing of leaves. Then, reductions in the rate of leaf and root growth are also due to salinity-induced water stress [11]. Similarly, in our study, the acceleration of leaf yellowing and the inhibition of water content and root development could be observed, which further confirmed the detrimental effect of salinity on plant seedling growth and development by regulating water limitation and growth reduction. Additionally, salt stress-induced osmotic stress severely reduces the root capability to uptake water and nutrients in soils. Moreover, the architecture and redox status of the root system undergoes various changes under salt stress [16,17]. Therefore, the reduction in root activity of seedlings under various NaCl stress could be accelerated through inhibition of water or nutrient absorption by roots and the changes of root-system architecture.

MAPK signaling occurs in response to almost any change in the extracellular or intracellular milieu via phosphorylation of downstream signaling targets, thereby regulating the metabolism of the cell, organ or the entire organism [18]. Interestingly, MAPK protein kinases also affect various intracellular responses and functions in inflammation, cell death, cell-cycle regulation, differentiation, development, senescence and tumorigenesis [19]. MAPK components are a set of enzymes causing plant response to the stimuli induced by different stress conditions [18,20]. In our study, RNA-seq results revealed that 42 DEGs were significantly enriched in the MAPK signaling pathway (sly04016) in tomato seedlings under salt stress (Figure 4 and Appendix A). In addition, the MAPK signaling pathway was mainly involved in plant hormone, hydrogen peroxide (H_2_O_2_), wounding and pathogen infection signaling pathways. Huang et al. [21] revealed the role of MAPK signaling in cell wall maintenance in a *Saccharomyces cerevisiae* FKS1 mutant by RNA-Seq analysis of transcriptomic changes. Recently, the crosstalk mechanisms between MAPK cascades and plant hormone were highlighted in plants, including auxin, ETH, ABA, JA, salicylic acid and brassinosteroid [22]. In our study, DEGs related plant hormone ETH, ABA and JA were identified in the MAPK signal pathway during salt stress (Figure 4). ETH is a gaseous hormone involved in various aspects of plant biology, including germination, plant growth, organ senescence and fruit ripening [23,24]. In our study, 11 DEGs related to ETH signaling in the MAPK pathway, including *ETH receptor*, *ETR*, *EIN/**EIL*, ethylene response factor *ERF* and *CHI* were regulated under salt stress. Analogously, the overexpression of *ERFs* enhanced the resistance of salinity via activating MAPK signaling cascade in rice [25] and tobacco [26]. High chitinase activities have been associated with stress tolerance in different plant species [27]. A chitinase-like protein was induced in NaCl-stress adaptive croton *Stellatopilosus Ohba* callus [28], and *chitinase* (*ChI11*) genes have been reported to confront salt stress tolerance in tomato [29]. These results implied that ETH-related genes enriched in MAPK signaling were involved in the resistance to salt stress. Interestingly, we found that salt stress could promote ETH production and this promotion was reversed by MAPK inhibitor SB (Figure 5), implying the essential roles of MAPK in salt-induced ETH production. Kim et al. [30] indicated that *NtMPK6-1* could induce ETH production. In addition, the treatment of dexamethasone induced the immediate activation of *NtMPK6-1* with a marked increase in ETH production [22]. Moreover, wounding-caused ETH production was also under the control of MAPKs [31]. As a consequence, one of the receptor-like kinases (RLKs) related to salt tolerance could phosphorylate both MPK3 and MPK6, implying that the resistance of salt stress is able to depend on the activation of MAPK signal [32]. Thus, the results mentioned above highlight that MAPK could be involved in salt stress-induced activation of ETH signaling and biosynthesis in tomato seedlings. The core ABA signaling is composed of PYR/PYL/RCAR receptors, PP2C phosphatases and SnRK2 kinases. We found that salt stress downregulated *PP2C*, *PP2C 53*, *PP2C 37* and *PYL3*, and upregulated *PP2C 51-like* and the serine/threonine-protein kinase gene *SRK2I* (Figure 3 and Figure 4). Arshad and Mattsson [33] indicated that a putative poplar PP2C-encoding gene acted as a negative regulator of ABA responses and negatively regulated drought stress responses in transgenic *Arabidopsis thaliana*. In addition, *SnRKs* are positive regulators of ABA signaling [34]. This may be because, in the presence of ABA, the PYR/PYL/RCAR-PP2C complex formation results in the reduction of PP2C activity, thus activating SnRKs, which targets membrane proteins, ion channels and transcription factors, and accelerating transcription of ABA-responsive genes [34]. This implies that salt stress activates ABA signaling. Further, we found that salt stress could also increase ABA content, which was inhibited by MAPK inhibitor SB (Figure 5). Similarly, in peas, the selective MAPKK inhibitor PD98059 significantly decreased ABA-induced stomatal closure [35]. Moreover, an inhibitor of MAPK of SB could also block the ABA-mediated stomatal closure in *Vicia faba* [36]. This suggests that SB203580 might block ABA signaling and its biosynthesis under salt stress. Thus, these results indicated that MAPK activated ABA signaling with an increasing biosynthesis in response to salt stress. Plant hormone, JA, is an important regulator of plant growth and development and stress response. We found that salt stress reduced JA content and JA-related genes expression, which were further decreased by MAPK inhibitor SB (Figure 4 and Figure 5). Previously, tomato *SlMPK6-1* and *SlMPK6-2* acted as positive regulators of JA biosynthesis and signaling, and silencing of *SlMPK6-1* and *SlMPK6-2* decreased JA biosynthesis and expression levels of JA-related genes [37]. That is to say, MAPK is involved in salt stress inhibited JA signaling and biosynthesis. In general, MAPKs have an indispensable role in the regulation of hormone balance in tomato seedlings when responding to salt stress.

The overaccumulation of toxic ROS could disrupt the process of plant growth and development. H_2_O_2_ and O^2−^ are considered as major indicators of ROS [38]. During environmental stress of plants (e.g., UV, salt toxicity, wounding), ROS levels can increase dramatically [39]. In this study, we observed that H_2_O_2_ content and O^2−^ production rate increased progressively under salt stress, which was reversed by MAPK inhibitor SB (Figure 6). This further contributes to understanding the effects of MAPK on salt stress, which might be connected with MAPK-mediated ROS metabolism. Farrukh et al. [40] showed that UVB potently induced MAPK activation and oxidative stress in human skin fibroblasts. Then, glycyrrhizic acid inhibited reactive ROS-mediated photodamage by blocking MAPK pathway in UV-B irradiated human skin fibroblasts. Besides, MAPK inhibitor SB blocked H_2_O_2_-mediated stomatal closure, indicating the involvement of MAPK in H_2_O_2_-mediated stomatal closure in *Vicia faba* [36]. MEKK1-MKK5-MPK6 mediates salt-induced expression of iron superoxide dismutase gene, which further promoted ROS production [41]. Therefore, these results indicated that MAPK participated in salt stress induced ROS accumulation and ROS acted downstream of MAPK pathway in the growth process of tomato seedlings. ROS accumulation is counteracted by scavenging enzymes to maintain its level. SOD, POD, CAT and APX are important antioxidant enzymes and play important roles in enhancing the tolerance to environmental stresses in plants [38]. We found that MAPK inhibitor SB significantly reversed the NaCl-caused decrease in CAT activity and enhancement of SOD, POD and APX activities (Figure 7). Previous studies indicated that H_2_O_2_ treatment induced the activation of a 46-KD MAPK, which enhanced the expression of antioxidant genes and the total activities of the antioxidant enzymes CAT, APX and SOD in the leaves of maize plants. Such enhancements were blocked by pretreatment with MAPK kinase inhibitors, suggesting that MAPK is involved in the antioxidant defense of maize [42]. Interestingly, in rat livers, age-dependent changes of the antioxidant system are accompanied by altered MAPK activation [43]. Accordingly, MAPK components functioned in activating the antioxidant enzymes resulting from salt stress, causing plant response to the stimuli.

Plants have specific defense mechanisms that are activated after sensing pathogens, initially on the cell surface. In addition, the MAPK cascade can regulate plant immunity via WRKY transcription factors [44]. The WRKY transcription factors family contains one or two DNA-binding domains, known as the WRKY domains, which consist of about 60 amino acids. WRKY transcription factors are recognized as one of the largest families of transcriptional regulators and are exclusively found in plants. Previous studies reported that WRKY transcription factors have various biological functions in plant abiotic stress responses, disease resistance, nutrient deprivation, seed and trichome development, senescence, and other hormone-regulated and developmental processes across various crop species [45]. Among them, WRKY33 and WRKY22 are pathogen-inducible transcription factors, which are regulated by MPK3/MPK6 and MPK4 cascade, respectively. Moreover, MPK4-MKS1-WRK33 is present in a nuclear localized complex. For instance, activation of MPK4 by bacterial elicitors caused MKS1 phosphorylation of WRKY33 in MPK4 complexes and then functioned in pathogen infection [46]. Thus, we analyzed whether MAPK cascade affects the resistance of tomato seedlings to pathogen infection under salt stress. In our study, expression levels of WRKY transcription factors (*SlWRKY 22*, *SlWRKY 33A* and *SlWRKY 33*), *MPK3/6*, *MPK4* and *MSK1* were significantly downregulated under salt stress, which was further decreased after the supplement of MAPK inhibitor SB (Figure 8). There was also a report that phosphorylation of a *WRKY33* by two pathogen-responsive MAPKs promotes phytoalexin biosynthesis, therefore driving the metabolic flow of camalexin production in Arabidopsis caused by pathogens [47]. Thus, this indicated that the resistance of tomato seedlings to pathogen infection was weakened under salt stress, which was deteriorated by the MAPK inhibitor, implying that the activation of MAPK signaling could enhance the plant immunity against pathogen infection under salt stress. Similarly, Gao et al. [48] found that transgenic tomatoes of *SlWRKY8* overexpression displayed a higher resistance to the pathogen *Pst.* DC3000 and increased the transcription levels of pathogen-related genes. Meanwhile, the transgenic plants have also showed the alleviated wilting phenotype, reduced ROS content and higher levels of antioxidant enzyme activities under salt stresses, indicating that *SlWRKY8* positively modulated plant immunity against pathogen infection as well as in plant responses to salt stresses. These results were consistent with our study. That is to say, MAPK is involved in plant immunity against pathogen infection of tomato seedlings under salt stress. It is interesting that MKS1 were shown to interact with the WRKY transcription factors WRKY33, and WRKY33 acts as a substrate of MPK4 [49]. It assumed that MKS1 may promote MPK4-modulated activation of defense by coupling the specific WRKY transcription factors. However, *OsMPK6* negatively regulated the disease resistance to bacterial pathogens in rice [50], which implies the difference of MAPK compounds in various plant species in response to stress stimulus-caused pathogen infection. Together, this work provides a better understanding about the molecular mechanisms involved in the activation of multiple signaling pathways, and the function of MAPK in different signaling pathways during salt stress, which provide initial insights into the response mechanism of tomato seedlings under salt stress.

## 4. Materials and Methods

### 4.1. Plant Materials and Assay Conditions

In our experiment, the tomato ‘Micro-Tom’ (*Solanum lycopersicum*) was used as material. First, the seeds were disinfected with 1% of NaClO and transferred to 1/2 Hoagland solution for 1 week after germination, then continued to be cultivated in Hoagland solution for 3 weeks. Then, the plant seedlings of uniform growth were collected for the following treatments with different concentrations for 1 week: firstly, NaCl (50, 100, 150, and 200 mM); secondly, NaCl (150 mM) + SB203580 (SB, MAPK phosphorylation specific inhibitor) and SB alone treatment, which were added to the Hogland solution, respectively. Seedlings treated with the Hogland solution adding no extra treatment served as the control. The experimental environment was kept at 16 h light at 250 μmol m^−2^ s^−1^ photons irradiance at 26 °C and 8 h dark at 20 °C, and 60% relative humidity. 

### 4.2. Biometric Parameters

The plant height was measured directly from the basal part to the growing point using a straightedge. The stem diameter was measured using a Vernier caliper. Three technical replicates were assayed for each treatment. 

### 4.3. Measurement of Physiological Indexes

The fresh weight and dry weight were measured by using the aboveground part of the tomato seedlings. The fresh weight was measured using an electronic balance (SQP, Sartorius, Shanghai, China), and after, samples were dried at 80 °C to constant weight to obtain its dry weight. Full tomato leaves were first weighed to determine the fresh weight (FW), and then weighed again to determine the turgid weight (TW) after they had been fully hydrated in deionised water for 1 day at 4 °C in the dark. Finally, they were placed in a drying oven and dried at 80 °C to constant weight to obtain their dry weight (DW). Leaf relative water content (LWC) = (FW − DW)/(TW − DW) × 100% [51]. The tomato roots were washed with deionized water and excised at 2 cm from the root tips for 14 h after OTC application. Then, the root activity was measured using the TTC method as described by Yan et al. [52] with some modification.

### 4.4. RNA Extraction, cDNA Library Construction and Sequencing

Total RNA was extracted using a TaKaRa MiniBEST Plant RNA Extraction Kit (TaKaRa) after treatment for 7 days. RNA quality and quantity were assessed using a NanoDrop spectrophotometer and an Agilent 2100 spectrophotometer, and more than 1 μg RNA samples were used for the cDNA library construction and Illumina sequencing, which were completed by Beijing Novogene Biological Information Technology Co., Ltd. (Beijing, China).

RNA-seq libraries were constructed according to the instructions of the Illumina Standard mRNAseq library preparation kit (Illumina, CA, USA). The RNA-seq libraries were sequenced on the Illumina HiSeq platform to generate 150-nt paired-end reads. Then raw data was filtered using cutadapt software and clean reads were mapped to the reference genome sequence of Tomato SL2.40 (ITAG2.3) by using TopHat. The values of fragments per kilobase of transcript per million reads for each biological replicate were determined using Cufflinks. 

### 4.5. Analysis of DEGs and Functional Annotation

The expression level of each gene was normalized by Fragments per kilobase of transcript per million fragments mapped (FPKM). DESeq was employed to analyze the DEGs between samples. DEGs were identified based on an absolute log 2 (fold change) > 0 and *p*-value < 0.05. All unigenes were annotated by BLASTx against seven public databases, including NR database, NT database, Pfam database, KOG database, Swiss-Prot database, KEGG database, and GO database under a threshold E-value of ≤10^−5^. GO enrichment and the KEGG pathway annotation were performed using BLAST2GO software and KOBAS software, respectively [53,54].

### 4.6. Quantitative Real-Time PCR (qRT-PCR) Assays

To validate the RNA-Seq results, 13 DEGs involved in the MAPK signaling pathway were selected and qRT-PCR assays were performed. Total RNA extraction, cDNA synthesis and qRT-PCR were conducted as described by Gao et al. [55]. All the primers used for RT-PCR were designed using prime 5 software (Primer Premier v5.0, PREMIER Biosoft, CA, USA), as shown in Appendix A.

### 4.7. Measurements of Endogenous Plant Hormones

The extraction was determined using previously reported methods with slight modifications [56]. Tomato leaves (0.5 g) were ground with a mortar and pestle in 3 mL of cooled 80% (*v*/*v*) methanol solution. Then, after incubation for 12 h at 4 °C, the extract was centrifuged at 8000 rpm for 15 min, which was repeated twice. The supernatant was merged, and the volume was raised to 10 mL with 80% pure methanol. Two milliliters of the supernatant were evaporated in a rotary vacuum at 38 °C for 4 h, then dissolved with 2 mL 80% cold pure methanol. Finally, the extract was filtered with a 0.22 μm filter for liquid chromatograph detection. The concentrations of ABA were measured by Quaternary gradient ultra-fast liquid chromatograph using Waters Acquity ARC 600-2998 equipped with the Symmetry-C18 column (4.6 × 250 mm, 5 μm). Jasmonic acid (JA) content was measured by liquid chromatography–mass spectrometry equipped with a C-18 column (2.1 × 50 mm, 1.8 μm, Agilent, Agilent Technologies Co. Ltd., Santa Clara, CA, USA) at a flow rate of 0.3 mL min^−1^. ETH production was determined according to the method of Wang et al. [57] with some modifications. The four fully expanded upper leaves beneath the growing tip of the plant were sampled and were then placed in a 795 mL desiccative airtight container and incubated at 25 °C for 12 h. 1 mL of headspace gas from each container was collected using a gas-tight hypodermic syringe and injected into a gas chromatograph (GC-17A, Shimadzu, Kyoto, Japan) immediately for the ETH concentration measurement. The gas chromatograph was equipped with a flame ionization detector and an activated alumina column.

### 4.8. Determination of Superoxide Anion and Hydrogen Peroxide

Hydrogen peroxide (H_2_O_2_) content and superoxide anion (O^2−^) production rate was analyzed as previously described by Wei et al. [38]. The content is shown in Figure 6. 

### 4.9. Determination of the Activities of Antioxidant Enzyme

The supernatant liquid used for the assays of enzyme activity was prepared according to the method described by Wei et al. [38] with slight modification. 

CAT activity was determined based on the disappearance of H_2_O_2_ by measuring the decrease in absorbance at 240 nm of a reaction mixture containing 50 mM phosphate buffer (pH 7.8), 0.1% (*v*/*v*) Triton X-100, 0.3% H_2_O_2_ and 0.1 mL of enzyme extract in a 3 mL volume.

For superoxide dismutase (SOD) activity, the supernatant (20 μL) was added to 3 mL of substrate mixture containing the reaction mixture [1.5 mL phosphate buffer (0.05 M, pH 7.8), 0.3 mL methionine (130 mM), 0.3 mL nitroblue tetrazolium (750 μM), 0.3 mL ethylene diamine tetra-acetic acid (1 mM), 0.3 mL riboflavin (20 μM) and 0.3 mL distilled water] and was incubated at 30 °C for 30 min. The absorbance was monitored at 560 nm. One unit of SOD activity was defined as the amount of enzyme eliciting 50% inhibition of nitroblue tetrazolium photoreduction.

Peroxidase (POD) activity was measured from the change of absorbance at 470 nm caused by the oxidation of guaiacol. The supernatant (100 μL) was combined with 2.6 mL 0.3% guaiacol and 0.3 mL 0.6% hydrogen peroxide, and the optical opening was quantified by measuring the absorbance at 470 nm after minutes.

Ascorbate peroxidase (APX) activity was assayed by monitoring the rate of ascorbate oxidation at 290 nm. The assay mixture contained 5 mM AsA, 20 mM H_2_O_2_, 0.1 mM EDTA, 50 mM phosphate buffer (pH 7.0) and 0.1 mL of enzyme extract in a 3 mL volume.

### 4.10. Statistical Analysis

The data in this study were analyzed using Excel 2010 and SPSS 22.0 software (SPPS Inc., Chicago, IL, USA). Values were the means ± SE of at least three independent experiments. For statistical analysis, Duncan’s multiple range test (*p* < 0.05) was chosen to determine the significance of the results among different treatments.

## 5. Conclusions

In summary, the study revealed that NaCl might have a concentration-dependent effect on the growth and development of tomato. The transcriptome analysis identified a large number of candidate genes involved in the process of salt stress through pairwise comparisons. Especially, results indicated a mechanism of tomato seedlings in response to salt stress that involves a major pathway, the MAPK signaling pathway, which mainly includes plant hormone (ETH, ABA and JA), H_2_O_2_, wounding and pathogen infection. Meanwhile, MAPK was found to be involved in the regulation of hormone balance, ROS metabolism, antioxidant capacity and plant immunity to pathogen infection of tomato seedlings under salt stress. Collectively, this work provides a better understanding of the molecular mechanisms involved in the activation of multiple signaling pathways and the function of MAPK in different signaling pathways during salt stress, which provides initial insights into the response mechanism of tomato seedlings under salt stress. Such details can be used for the production of plants with a higher tolerance to such stresses.

## Figures and Tables

**Figure 1 ijms-23-07645-f001:**
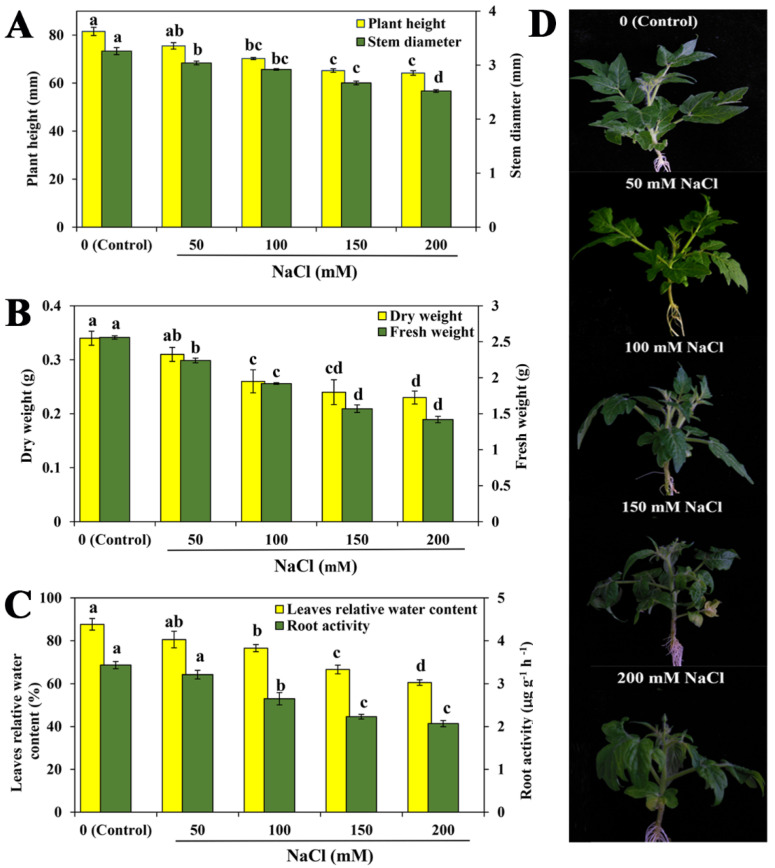
Effects of different concentrations of NaCl on tomato seedling growth. (**A**) Plant height and stem diameter. (**B**) Dry weight and fresh weight. (**C**) Leaf relative water content (LWC) and root activity. Photographs (**D**) were taken after seven days of the treatments indicated. The values (means ± SE) are the averages of three independent experiments (*n* = 15). Bars not sharing the same letters indicate statistically significant differences according to Duncan’s multiple range test (*p* < 0.05).

**Figure 2 ijms-23-07645-f002:**
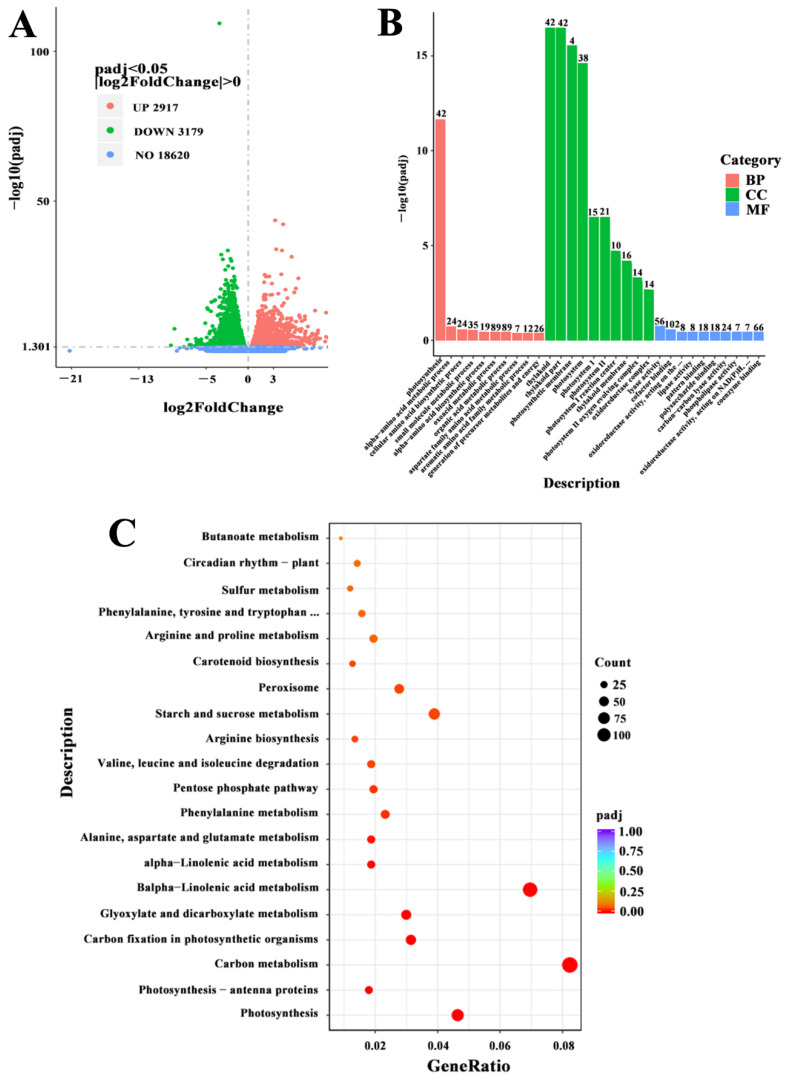
Transcriptome profiling analysis during salt stress. (**A**) The volcano plot showed the numbers of significantly differentially expressed genes (DEGs). Red points represent upregulated genes, green points represent downregulated genes, and blue points represent no differences. (**B**) Functional gene ontology (GO) term classifications of DEGs. GO terms were summarized in three main categories of cellular component, molecular function, and biological process. (**C**) The DEGs enrichment in different KEGG pathways. vs.: Versus.

**Figure 3 ijms-23-07645-f003:**
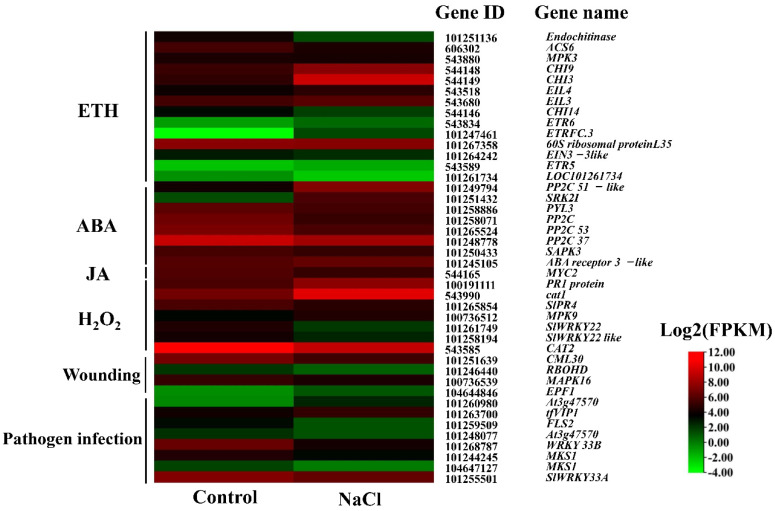
Expression patterns of differentially expressed genes related to MAPK signaling pathway during salt stress. The color scale corresponds to log^2−^ transformed (fragments per kilobyte per million reads) FPKM values with red indicating upregulation and green indicating downregulation. Each row represents a unigene. Control and NaCl represented different treatments.

**Figure 4 ijms-23-07645-f004:**
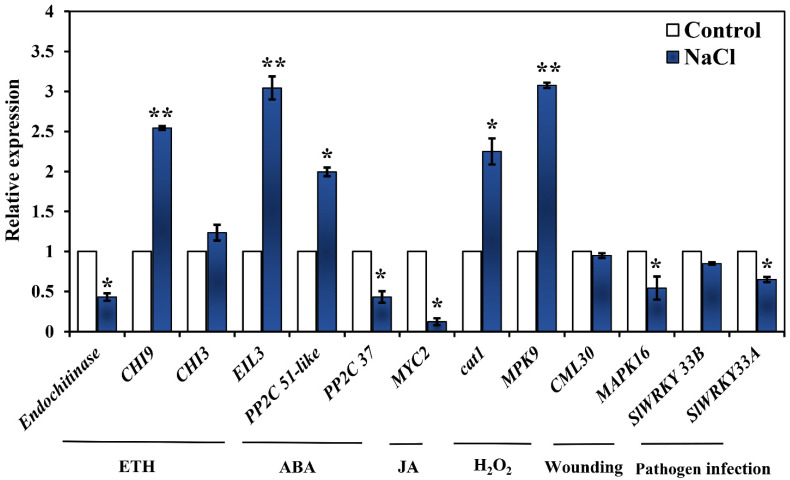
Quantitative real-time PCR analysis of the expression profiles of thirteen genes related to MAPK signal transduction during salt stress. The error line is the standard error. * and ** indicate significant differences (respectively, *p* < 0.05 and *p* < 0.01) according to Duncan’s multiple range test compared with the control.

**Figure 5 ijms-23-07645-f005:**
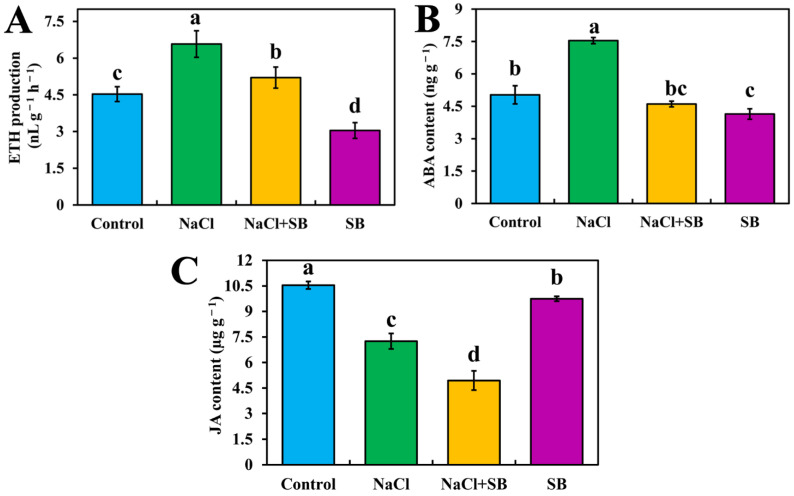
Effect of MAPK inhibitors SB203580 on the levels of endogenous hormones during salt stress. (**A**) ETH production, (**B**) ABA content, (**C**) JA content. The values (means ± SE) are the averages of three independent experiments (*n* = 15). Bars not sharing the same letters indicate statistically significant differences according to Duncan’s multiple range test (*p* < 0.05).

**Figure 6 ijms-23-07645-f006:**
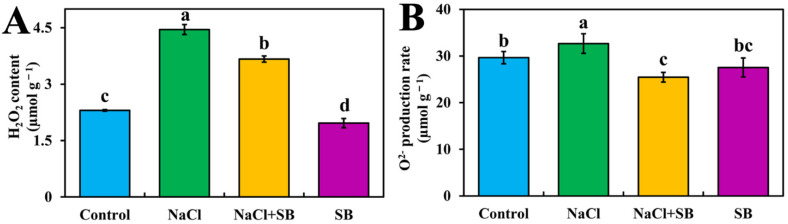
Effect of MAPK inhibitors SB203580 on H_2_O_2_ and O^2−^ content during salt stress. (**A**) H_2_O_2_ content, (**B**) O^2−^ content. The values (means ± SE) are the averages of three independent experiments (*n* = 15). Bars not sharing the same letters indicate statistically significant differences according to Duncan’s multiple range test (*p* < 0.05). DAB staining, NBT staining.

**Figure 7 ijms-23-07645-f007:**
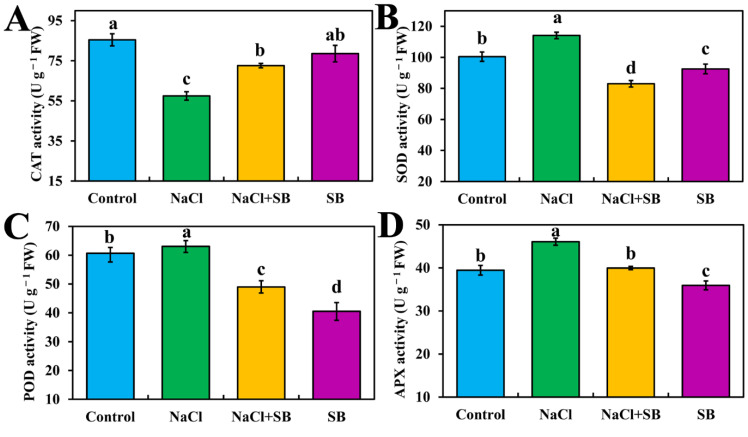
Effect of MAPK inhibitors SB203580 on the activities of antioxidant enzyme during salt stress. (**A**) CAT activity, (**B**) SOD activity, (**C**) POD activity, (**D**) APX activity. The values (means ± SE) are the averages of three-independent experiments (*n* = 15). Bars not sharing the same letters indicate statistically significant differences according to Duncan’s multiple range test (*p* < 0.05).

**Figure 8 ijms-23-07645-f008:**
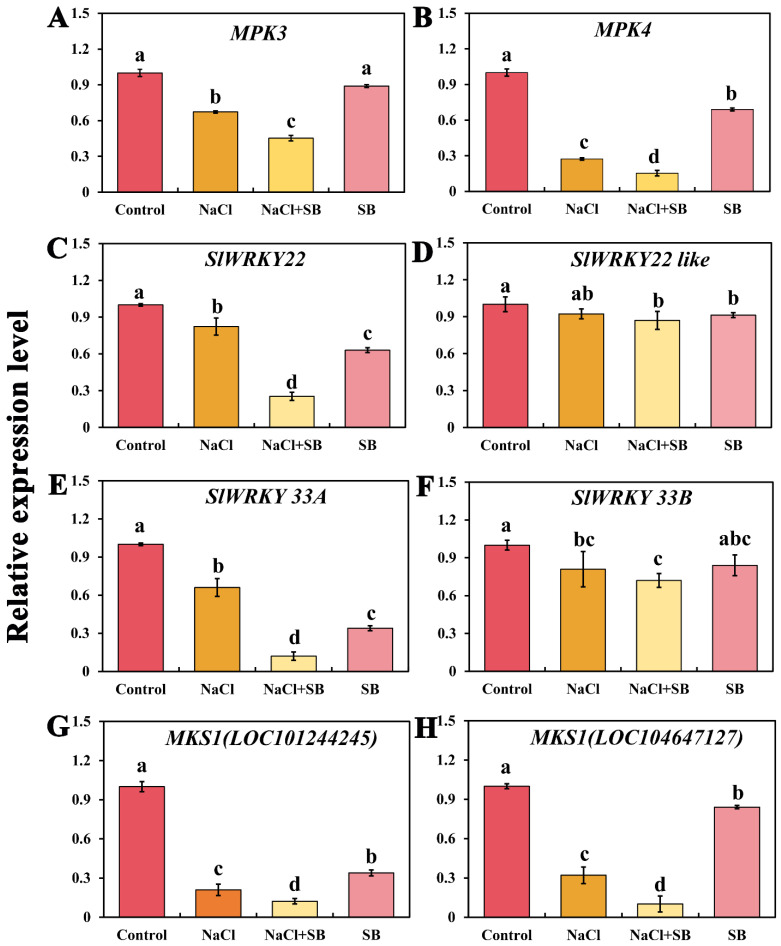
Effect of MAPK inhibitors SB203580 on the relative expression levels of genes related to defense response to pathogens. (**A**) *MPK3* expression level, (**B**) *MPK4* expression level, (**C**) *SlWRKY22* expression level, (**D**) *SlWRKY22 like* expression level, (**E**) *SlWRKY 33A* expression level, (**F**) *SlWRKY 33B* expression level, (**G**) *MSK1 (LOC 101244245)* expression level, (**H**) *MSK1* (*LOC 104647127)* expression level. The values (means ± SE) are the averages of three independent experiments (*n* = 15). Bars not sharing the same letters indicate statistically significant differences according to Duncan’s multiple range test (*p* < 0.05).

## Data Availability

Data are contained within the article or Appendix A.

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
