# Peer review of "Mitogen-Activated Protein Kinase Is Involved in Salt Stress Response in Tomato (Solanum lycopersicum) Seedlings"

_ijms, 2022, doi:10.3390/ijms23147645_

Round 1

Reviewer 1 Report

The text bellow contains comments on manuscript entitled “Mitogen-activated protein kinase is involved in salt stress response in tomato (Solanum lycopersicum) seedlings”.

The manuscript is focused on salinity response investigations over the activities of the antioxidant enzymes in tomato seedlings. Differentially expressed genes (DEGs) were identified and classified into different metabolic pathways, including photosynthesis, carbon metabolism, biosynthesis of amino acids and mitogen-activated protein kinase (MAPK) signaling pathway.

The manuscript is well written with a logically structured experimental design and results and discussion supporting the aim of the study.

I have few recommendations, which the authors might consider if they find them appropriate.

I believe that the authors should be more precise in formulating the aim of their study and the expected outcomes. Is their approach unique/applied for first time for example?

I think that the authors should provide figures with a better quality and consistency in font style and size.

Author Response

Response to Reviewer #1

Comments:

The text bellow contains comments on manuscript entitled “Mitogen-activated protein kinase is involved in salt stress response in tomato (Solanum lycopersicum) seedlings”.

The manuscript is focused on salinity response investigations over the activities of the antioxidant enzymes in tomato seedlings. Differentially expressed genes (DEGs) were identified and classified into different metabolic pathways, including photosynthesis, carbon metabolism, biosynthesis of amino acids and mitogen-activated protein kinase (MAPK) signaling pathway.

The manuscript is well written with a logically structured experimental design and results and discussion supporting the aim of the study.

I have few recommendations, which the authors might consider if they find them appropriate.

  • I believe that the authors should be more precise in formulating the aim of their study and the expected outcomes. Is their approach unique/applied for first time for example?
  • I think that the authors should provide figures with a better quality and consistency in font style and size.

Response:

We would like to thank the reviewer for the thoughtful comments and constructive suggestions.

  • Thank you for your valuable comments. According to your suggestions, we have revised the aim of study and the expected outcomes and made them more precise in discussion part. (More details please see the revised manuscript). The related statements are as follows:

Furthermore, there are a lot of missing links between tomato seedling growth and MAPKs under salt stress. Therefore, it is important to discover the possible effects of various MAPK signaling on growth and development in tomato seedlings under sodium chloride (NaCl) stress and their possible mechanisms. Here, our objectives of the present study were to investigate the impact of NaCl as moderate salt stress on the physio-morphological attributes and the growth of tomato cv. ‘Micro-Tom’ seedlings. Then, RNA sequencing (RNA-seq) technology was conducted to identify the candidate genes associated with MAPK signaling pathway between salt stressed-tomato seedlings and the control. Last, according to RNA-seq results associated with MAPK signaling pathway, we mainly and firstly focused on the change of the levels of endogenous hormones, ROS metabolism, antioxidant ability and defense response for pathogen after supplying MAPK inhibitors SB203580 in tomato seedlings under salt stress.  

  • We have revised all figures and made them better quality and consistency in manuscript, including font style and size, resolution and color matching of figures. (More details please see the revised figures in revision).

Reviewer 2 Report

Minor comments

1) At the end of the Introduction section the authors write: ‘this work provides a better understanding about the molecular mechanisms about the activation of multiple signaling pathways, and the function of MAPK  (Mitogen-activated protein kinases)   in different signaling pathways during salt stress, which provide initial insights into the response mechanism of tomato seedlings under salt stress’- this is a conclusion and I wonder whether it should be written at the end of Introduction

2)section 2.1. Decipher LWC in the text and in Fig.1 legend

3)section 2.2. Decipher DEG (Page 3)

4) GO should be deciphered in the text not only in Figure 2 legend

Author Response

Response to Reviewer #2

Comments

1) At the end of the Introduction section the authors write: ‘this work provides a better understanding about the molecular mechanisms about the activation of multiple signaling pathways, and the function of MAPK (Mitogen-activated protein kinases) in different signaling pathways during salt stress, which provide initial insights into the response mechanism of tomato seedlings under salt stress’- this is a conclusion and I wonder whether it should be written at the end of Introduction

2)section 2.1. Decipher LWC in the text and in Fig.1 legend

3)section 2.2. Decipher DEG (Page 3)

4) GO should be deciphered in the text not only in Figure 2 legend

Response:

Thank you very much for your comments!

  • We have removed this sentence “this work provides a better understanding about the molecular mechanisms about the activation of multiple signaling pathways, and the function of MAPK (Mitogen-activated protein kinases) in different signaling pathways during salt stress, which provide initial insights into the response mechanism of tomato seedlings under salt stress” from introduction part to conclusion part. (More details please see the revised manuscript).
  • The explanation of LWC “leaf relative water content” has been added in the text and in Fig.1 legend. (More details please see the revised manuscript).
  • The explanation of DEG “differentially expressed gene” has been added in the text and in Fig.1 legend. (More details please see the revised manuscript).
  • The explanation of GO “gene ontology” has been added in the text. (More details please see the revised manuscript).
